# NUTS: Eddy-Robust Reconstruction of Surface Ocean Nutrients via Two-Scale Modeling

**Hao Zheng**[1][\*]    **Shiyu Liang**[1][\*][†] **Yuting Zheng**[1]    **Chaofan Sun**[1]    **Lei Bai**[2]    **Enhui Liao**[1]

[1]Shanghai Jiao Tong University, China    [2]Shanghai Artificial Intelligence Laboratory, China

`{hubert.zheng, lsy18602808513, zhengyt058, scf024, ehliao}@sjtu.edu.cn`
`baisanshi@gmail.com`

## Abstract

Reconstructing ocean surface nutrients from sparse observations is critical for understanding long-term biogeochemical cycles. Most prior work focuses on reconstructing atmospheric fields and treats the reconstruction problem as image inpainting, assuming smooth, single-scale dynamics. In contrast, nutrient transport follows advection–diffusion dynamics under nonstationary, multiscale ocean flow. This mismatch leads to instability, as small errors in unresolved eddies can propagate through time and distort nutrient predictions. To address this, we introduce NUTS, a two-scale reconstruction model that decouples large-scale transport and mesoscale variability. The homogenized solver captures stable, coarse-scale advection under filtered flow. A refinement module then restores mesoscale detail conditioned on the residual eddy field. NUTS is stable, interpretable, and robust to mesoscale perturbations, with theoretical guarantees from homogenization theory. NUTS outperforms all data-driven baselines in global reconstruction and achieves site-wise accuracy comparable to numerical models. On real observations, NUTS reduces NRMSE by 79.9% for phosphate and 19.3% for nitrate over the best baseline. Ablation studies validate the effectiveness of each module.

## 1 Introduction

Reconstructing historical nutrient concentrations in the surface ocean is essential for understanding long-term biogeochemical cycles, ecosystem variability, and anthropogenic influence Stüeken et al. [2024]. However, nutrient observations are extremely sparse, especially before the bio-Argo era when data came from irregular ship-based campaigns. Even today, nutrient data remain far less available than satellite-measured variables like sea surface temperature (SST) or chlorophyll Mishonov et al. [2024], Locarnini et al. [2018].

Recent deep learning advances have driven progress in forecasting and reconstructing high-dimensional atmospheric fields. Transformer-based models Pathak et al. [2022], Bi et al. [2023], Lam et al. [2023] achieve state-of-the-art short-term forecasts by capturing temporal dependencies in data-rich regimes with complete initial conditions. In contrast, climate field reconstruction operates in sparse settings and is often framed as a spatial in-painting task Bochow et al. [2025], Plésiat et al. [2024], Kadow et al. [2020]. Early models Ronneberger et al. [2015], Dosovitskiy et al. [2020], Gao et al. [2022] focus on spatial correlations, while recent hybrids Li et al. [2020], Wang et al. [2025], Beauchamp et al. [2023] add physical constraints for greater consistency. However, these methods are mainly designed for smooth, single-scale wind fields with well-resolved large-scale structure.

---

[\*]Equal contribution.
[†]Corresponding author.

Reconstructing ocean nutrients demands a fundamentally different approach. Unlike atmospheric fields, nutrient transport follows advection–diffusion dynamics driven by a nonstationary, multiscale velocity field. Surface currents consist of a slowly evolving large-scale mean flow overlaid with rapidly fluctuating mesoscale eddies. These eddies—coherent vortices spanning 10–100 km—govern most lateral nutrient transport Vallis [2017], McWilliams [2016], Chelton et al. [2011], yet are poorly resolved in numerical circulation models due to limited resolution and inherent uncertainty. As a result, reconstruction models that rely directly on such flow fields are fragile: even small perturbations in the eddy component can degrade predictions.

Robust nutrient reconstruction presents a core modeling dilemma. Filtering the input velocity field improves stability by suppressing high-frequency eddy perturbations. However, it also removes fine-scale structures essential for capturing local nutrient gradients. Retaining all scales introduces instability; over-filtering sacrifices resolution. A principled solution must separate scales—preserving large-scale transport while reintroducing mesoscale variability in a controlled manner.

We propose NUTS, a novel and robust two-scale model that, for the first time, resolves the reconstruction challenge through a structured decomposition. At its core is a *homogenized advection–diffusion solver* that models nutrient transport under the filtered large-scale flow. By replacing unresolved mesoscale variability with an effective diffusion term, this formulation captures the net impact of fine-scale dynamics without tracking unstable eddy fluctuations. The *coarse module* leverages this framework to propagate nutrient fields with stability and physical consistency. To recover fine-scale structure, the *refinement module* models localized redistribution conditioned on the residual mesoscale flow and the coarse prediction. This coarse-to-refined architecture preserves large-scale transport patterns while restoring spatial detail in dynamically active regions. NUTS is robust to mesoscale perturbations, respects scale separation, and generalizes effectively under sparse observational coverage. We establish accuracy and stability guarantees under standard assumptions from homogenization theory, and empirically demonstrate that NUTS consistently outperforms prior baselines on both simulated and real-world datasets. Our contributions are as follows:

- We formulate nutrient reconstruction as a spatiotemporal advection–diffusion problem and reveal the vulnerability of naive methods to mesoscale perturbations.

- We propose NUTS, a two-scale model that combines a homogenized PDE solver with adaptive diffusion and a refinement module conditioned on normalized eddy flow. We provide theoretical justification of its effectiveness under standard homogenization assumptions.

- We empirically demonstrate that NUTS outperforms all data-driven baselines in global nutrient reconstruction on both simulated and real-world datasets, achieving site-wise accuracy comparable to physics-based numerical models. On the WOD dataset of real observations, NUTS reduces NRMSE by 79.9% for phosphate and 19.3% for nitrate relative to the best baseline.

- Ablation studies highlight the contribution of each component and offer empirical guidance for designing robust reconstruction architectures.

## 2 Related Work

This section outlines key related work and we provide a comprehensive review with extended background and references in Appendix A.

**Nutrient Data.** Ocean nutrient data are typically derived from observational datasets and simulation-based products. Raw observational archive, such as WOD Mishonov et al. [2024], provide high-quality and in-situ measurements but suffer from sparse and uneven distribution. In contrast, simulation-based products like the CMEMS Global Ocean Biogeochemical Hindcast (GOBH) Perruche [2018], ECCO-Darwin Carroll et al. [2020], MOM6-COBALT2 Griffies et al. [2012] can offer global coverage data product with coupled physical and biogeochemical dynamics, but require extensive calibration. Furthermore, most of these simulation-based products do not incorporate biogeochemical data assimilation and often employ simplified parameterizaton of biogeochemical processes, resulting in regional biases and uncertainties in nutrient fields.

**Reconstruction Approaches.** Traditional methods such as optimal interpolation Conkright et al. [2002], 3D/4D-Var Courtier et al. [1994], ensemble Kalman filters Nerger and Gregg [2008], and variational inverse models Brasseur and Haus [1991] rely on data assimilation and inverse modeling to integrate sparse observations with physical dynamics, but are often limited by computational cost

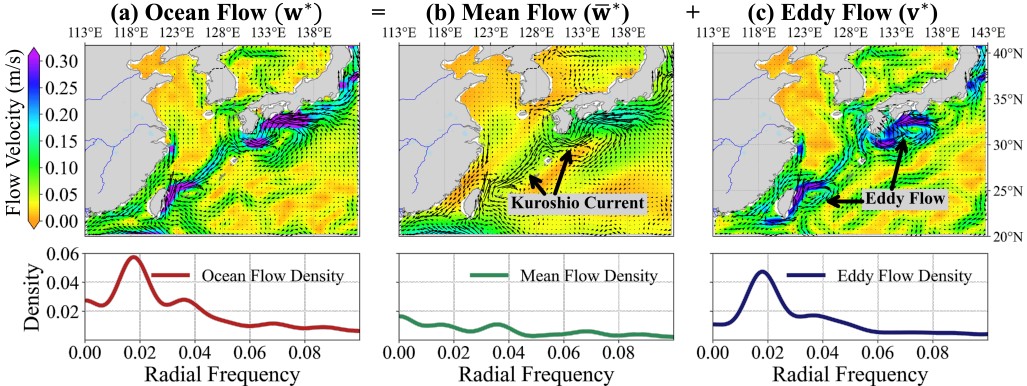

Figure 1: **Decomposition of surface ocean flow into mean and eddy components.** **(a)** Full velocity field containing both large-scale and mesoscale structures. **(b)** Mean flow obtained via low-pass filtering, capturing large-scale structures such as the Kuroshio Current. **(c)** Eddy flow computed as the residual, representing high-frequency mesoscale variability. Bottom panels display the radial frequency spectra corresponding to each flow component, with energy concentrated at low frequencies for the mean flow and at higher frequencies for the eddy flow, illustrating effective scale separation.

and data sparsity. Recent advances in deep learning offer alternative solutions for spatiotemporal reconstruction. CNN-based models (e.g., U-Net Ronneberger et al. [2015]) and transformers (e.g., ViT Dosovitskiy et al. [2020], Earthformer Gao et al. [2022]) capture spatial structures but lack physical grounding. Physics-informed approaches—such as neural operators Li et al. [2020], Wang et al. [2025], implicit neural representations Luo et al. [2024], and 4DVarNet Beauchamp et al. [2023]—embed governing equations or physical constraints into the learning process to improve physical consistency but are limited in capacity. Foundation models (e.g., Prithvi Schmude et al. [2024], AtmoRep Lessig et al. [2023]) show promise in meteorology but remain untested in marine biogeochemistry. General-purpose inpainting methods using GANs Zhao et al. [2021] and diffusion models Lugmayr et al. [2022] perform well in vision tasks but lack physical constraints and robustness to sparse data.

## 3  Methodology

**Notations.** Let $\mathbb{S}^2$ denote the unit surface in $\mathbb{R}^3$, parameterized by latitude-longitude coordinates $\mathbf{x} = (\theta, \phi) \in \Omega = [-\frac{\pi}{2}, \frac{\pi}{2}] \times [-\pi, \pi]$. For a time-dependent function $\varphi(\theta, \phi, t)$, define $\dot{\varphi} = \frac{\partial \varphi}{\partial t}$. The divergence and spherical Laplacian operators are denoted $\nabla \cdot$ and $\nabla^2$, respectively.

**Problem Setup.** The nutrient concentration $\varphi$ follow the advection-diffusion equation: $\mathcal{L}_{\mathbf{w}, \eta}[\varphi] = \dot{\varphi} + \nabla \cdot (\mathbf{w}\varphi) - \eta\nabla^2\varphi = s$, where $\eta = \eta(\theta, \phi)$ denotes the time-invariant diffusion coefficient and $s = s(\theta, \phi, t)$ represents the external source and sink terms. These terms account for biological uptake and remineralization through photosynthesis, respiration, and demineralization, as well as physical downwelling and upwelling. Given sparse nutrient measurements on $\mathcal{Z} \times \mathcal{T} \subset \Omega \times [0, T]$ and perturbed ocean flow estimates $\mathbf{w}$, our goal is reconstructing nutrient concentrations by solving the constrained PDE: $\mathcal{L}_{\mathbf{w}, \eta}[\varphi] = s$, subject to $\varphi|_{\mathcal{Z} \times \mathcal{T}} = f|_{\mathcal{Z} \times \mathcal{T}}$, where $f$ represents observed nutrient concentrations and $f|_{\mathcal{Z} \times \mathcal{T}}$ denotes its restriction to the subset $\mathcal{Z} \times \mathcal{T}$.

### 3.1  A Naive Spatial-Temporal Reconstruction Model

In this subsection, we introduce a naive spatiotemporal reconstruction model and discuss its advantage over image-inpainting-based methods. We then demonstrate its sensitivity to mesoscale perturbations in the eddy component of the ocean velocity field.

**Naive Model.** Given an interval $[t_0, t_1]$, the naive model first uses a data-driven initializer $\mathcal{F}_0$ to estimate the initial nutrient field $\hat{\varphi}(\mathbf{x}, t_0)$ through the velocity field $\mathbf{w}$, auxiliary variables $\mathbf{\Phi}$, and sparse observations $f$. The estimate is then propagated by solving the advection–diffusion equation $\mathcal{L}_{\mathbf{w}, \eta}[\hat{\varphi}] = s$, where both the diffusion coefficient $\eta$ and source term $s$ are learned to match the true field. Prior work Schiesser [2012] has demonstrated that this propagation can be implemented via the method of lines (MOL), which discretizes the PDE into a system of first-order ODEs at spatial locations $\{\mathbf{x}_k\}_k$: $\hat{\varphi}(\mathbf{x}_k, t) = \hat{\varphi}(\mathbf{x}_k, t_0) + \int_{t_0}^{t} \left[ -\nabla \cdot (\mathbf{w}\varphi) + \eta\nabla^2\varphi + s \right] (\mathbf{x}_k, \tau)\mathrm{d}\tau$, where the forward solution can be solved approximately using numerical solvers such as Runge–Kutta LeVeque [2007]. During the model training, all components, i.e., $\mathcal{F}_0$, $s$ and $\eta$ are jointly optimized to minimize

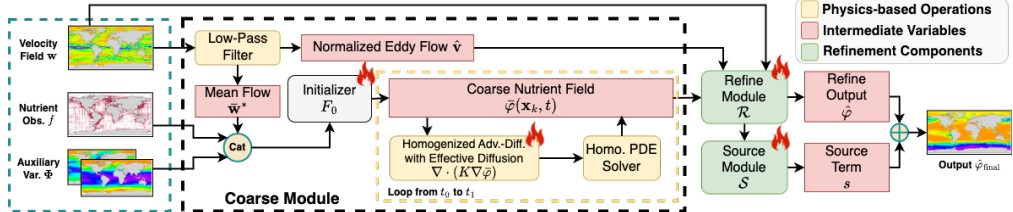

Figure 2: **Overview of NUTS.** NUTS is a two-scale model that combines a data-driven initializer, a homogenized PDE solver with learned effective diffusion, and a refinement module to reconstruct ocean nutrients under sparse observations. The velocity field is decomposed into mean and eddy components for scale separation. A learnable source module captures unresolved inputs. Trainable modules are marked with fire icons. Cat denotes channel concatenation and $\oplus$ denotes element-wise addition.

the mean squared error between the prediction $\hat{\varphi}$ and the ground-truth $\varphi$, i.e., $\min_{\mathcal{F}_0, s, \eta} \mathcal{L}_{\text{MSE}} \triangleq \|\varphi - \hat{\varphi}\|_2^2$.

**Advantages over Existing Image In-painting Approach.** (1) *Physical consistency and mass conservation.* The naive model evolves nutrient fields through an advection–diffusion PDE, ensuring temporally consistent reconstructions that follow physical transport processes and conserve mass. In contrast, image in-painting methods rely purely on spatial interpolation, lacking temporal dynamics and physical grounding. (2) *Effective use of sparse observations.* By jointly learning the initializer, source term, and diffusion coefficient, the naive model directly integrates observational data to constrain transport dynamics, leading to more data-consistent estimates in sparsely sampled regions.

**Sensitivity to Mesoscale Perturbations.** The naive reconstruction approach evolves nutrient estimates using velocity fields from numerical circulation models, which accurately capture large-scale mean currents but often misrepresent mesoscale eddies due to limited resolution and structural uncertainties. Mesoscale eddies are small-scale (10–100 km), high-energy structures that play a dominant role in nutrient transport. As illustrated in Figure 1, the true velocity field $\mathbf{w}^*$ can be decomposed into a smooth mean component $\bar{\mathbf{w}}^*$ and a rapidly fluctuating eddy component $\mathbf{v}^*$. When the MOL uses a perturbed velocity field $\mathbf{w}_\delta = \mathbf{w}^* + \boldsymbol{\delta}$, structural errors $\boldsymbol{\delta}$ in the eddy component introduce an additional transport term into the advection–diffusion dynamics:

$$\hat{\varphi}(\mathbf{x}_k, t_1) = \hat{\varphi}(\mathbf{x}_k, t_0) + \int_{t_0}^{t_1} \underbrace{\left[ -\nabla \cdot (\mathbf{w}^* \varphi) + \eta \nabla^2 \varphi + s \right](\mathbf{x}_k, \tau)}_{\text{true advection–diffusion}} - \underbrace{\left[ \nabla \cdot (\boldsymbol{\delta} \varphi) \right](\mathbf{x}_k, \tau)}_{\text{error from flow perturbation}} \, d\tau. \quad (1)$$

The perturbation term scales with $\|\boldsymbol{\delta}\|$, which can be large, as mesoscale eddies typically carry more energy than the mean flow (Figure 1). By Equation (1), such perturbations induce significant transport errors that accumulate and propagate over time. This underscores the need for reconstruction models that are robust to mesoscale flow inaccuracies.

## 3.2 NUTS: Eddy-Robust Nutrient Reconstruction via Two-Scale Modeling

We introduce **NUTS**, a principled two-scale model that reconstructs surface ocean nutrients from sparse observations and noisy velocity inputs (see Figure 2). Unlike prior approaches, NUTS separates nutrient transport into stable mean dynamics and unstable mesoscale variability. It applies a homogenized PDE solver for large-scale propagation and a refinement module for controlled recovery of fine-scale structure. This decomposition improves robustness and generalization in multiscale ocean flows. All architectural details are provided in Appendix B.

**Coarse Module Part I: Robust Initializer.** The coarse stage begins by estimating the nutrient field $\bar{\varphi}(\mathbf{x}_k, t_0)$ at the start of the reconstruction interval. To suppress mesoscale noise, we apply a Fourier-based low-pass spatial filter to the input velocity field and extract the mean flow $\bar{\mathbf{w}}^*$. This filtered flow, along with sparse nutrient observations and auxiliary variables, is encoded by a spatiotemporal transformer that captures long-range dependencies across space and time. The initializer is designed to be robust to flow perturbations and produces a stable starting point for physical propagation.

**Coarse Module Part II: Homogenized PDE Solver.** To evolve the field forward, NUTS applies a homogenized advection–diffusion equation:

$$\bar{\varphi}(\mathbf{x}_k, t_1) = \bar{\varphi}(\mathbf{x}_k, t_0) + \int_{t_0}^{t_1} \left[ \underbrace{-\nabla \cdot (\bar{\mathbf{w}}^* \hat{\varphi})}_{\text{mean flow advection}} + \underbrace{\nabla \cdot (K \nabla \bar{\varphi})}_{\text{effective diffusion}} \right](\mathbf{x}_k, \tau) \, d\tau. \quad (2)$$

This formulation replaces unresolved mesoscale effects with an effective diffusion tensor $K(\mathbf{x})$, which is predicted by a hypernetwork conditioned on $\bar{\varphi}$. We discretize the system using the method of lines and numerically integrate it over time. This structured PDE solver ensures stable and physically grounded transport under filtered dynamics.

**Refinement Module.** The refinement stage corrects residual errors and restores mesoscale variability. It takes as input the coarse prediction $\bar{\varphi}$, mean flow $\bar{\mathbf{w}}^*$, normalized eddy velocity $\hat{\mathbf{v}} = (\mathbf{w} - \bar{\mathbf{w}}^*)/\|\mathbf{w} - \bar{\mathbf{w}}^*\|_\infty$, sparse observations, and static covariates. These inputs are tokenized and passed through a vision transformer $\mathcal{R}$ that produces the refined estimate $\hat{\varphi}(\mathbf{x}, t)$, i.e., $\hat{\varphi}(\mathbf{x}, t) = \mathcal{R}\left[\bar{\varphi}, \bar{\mathbf{w}}^*, \hat{\mathbf{v}}, \mathbf{\Phi}, f\right](\mathbf{x}, t)$. Refinement is performed independently at each timestep and learns localized spatial redistribution driven by eddy structures.

**Source Term and Conservation Loss.** To account for unresolved sources and sinks, we introduce a learnable correction term $s = \mathcal{S}(\hat{\varphi})$, where $\mathcal{S}$ is parameterized by a ResNet. The final prediction is $\hat{\varphi}_{\text{final}} = \hat{\varphi} + s$. To enforce physical realism, we define total nutrient mass as $M[\varphi](t) = \sum_k \varphi(\mathbf{x}_k, t)$, and penalize mass drift through the conservation loss:

$$\mathcal{L}_{\text{cons.}} = \int_{t_0}^{t_1} |M[\bar{\varphi}](\tau) - M[\bar{\varphi}](t_0)|^2 + |M[\hat{\varphi}](\tau) - M[\bar{\varphi}](t_0)|^2 \, d\tau.$$

The final training objective is: $\mathcal{L}_{\text{total}} = \|\hat{\varphi}_{\text{final}} - \varphi\|_2^2 + \lambda \mathcal{L}_{\text{cons.}}$, which governs the optimization of all learnable components in NUTS.

**Core Insight: Why Two-Scale Modeling Works.** The key challenge in reconstructing ocean nutrient fields lies in the dual nature of the underlying dynamics: large-scale currents govern basin-wide transport, while mesoscale eddies induce localized variability and dominate error sensitivity. NUTS addresses this by explicitly separating these two regimes. The coarse module filters out unstable mesoscale fluctuations and models stable transport via a homogenized PDE with learnable diffusion. This prevents error accumulation from uncertain eddy inputs. The refinement module then selectively reintroduces mesoscale information—not as direct forcing, but as spatial corrections conditioned on the residual flow. This two-stage architecture mirrors the physical structure of ocean transport and enables both robustness and resolution in a way that single-scale models cannot.

**Advantages over the Naive Approach.** NUTS preserves the physical grounding of the naive model, including advection–diffusion transport and the effective use of sparse observations. But it adds two critical improvements: (1) *Scale-aware architecture.* By decoupling mean and eddy-driven dynamics, NUTS reconstructs both broad circulation and localized nutrient features with greater fidelity. (2) *Built-in robustness.* Homogenization shields the system from mesoscale perturbation errors, while spatial refinement restores resolution without destabilizing temporal evolution.

**Context and Relation to Prior Work.** While prior hybrid models such as FNO Li et al. [2020], 4DVarNet Beauchamp et al. [2023], and GraphCast Lam et al. [2023] embed physical priors into data-driven forecasting pipelines, they typically rely on direct PDE application or learn-to-solve strategies that do not explicitly separate stable and unstable components. In contrast, NUTS reformulates the transport equation itself: it applies homogenization to eliminate mesoscale instability at the PDE level and delegates high-frequency recovery to a separate spatial refinement module. This scale-aware decomposition is essential for robustness in noisy flow regimes.

### 3.3 Theoretical Analysis: Effectiveness of NUTS under Eddy Perturbations

We adopt a standard multiscale formulation for ocean velocity Pavliotis and Stuart [2008], modeling $\mathbf{w}^*(\mathbf{x}, t) = \bar{\mathbf{w}}^*(\mathbf{x}, t) + \frac{1}{\varepsilon}\mathbf{v}^*(\mathbf{x}, t; \mathbf{y}, \tau)$, where $\varepsilon \ll 1$ characterizes the scale separation between slow large-scale transport and fast mesoscale variability, and $\mathbf{y} = \mathbf{x}/\varepsilon$, $\tau = t/\varepsilon^2$ are fast space-time variables that resolve high-frequency eddy dynamics. The mean flow $\bar{\mathbf{w}}^*$ governs large-scale advection, while $\mathbf{v}^*$ captures mesoscale eddies with rapid, oscillatory fluctuations. This parabolic scaling is standard in homogenization theory for advection–diffusion systems Pardoux and Veretennikov [2005], ensuring that mesoscale variability mixes locally without inducing net large-scale transport. We further assume that both $\mathbf{v}^*$ and the perturbation $\boldsymbol{\delta}$ satisfy the same structural form: periodic and mean-zero in the fast variables $(\mathbf{y}, \tau)$. This assumption is classical in homogenization theory Jikov et al. [2012] and reflects the physical behavior of mesoscale eddies—highly energetic but oscillatory and net-zero under space-time averaging.

**Theorem 1** (Informal; Accuracy and Robustness under Eddy Perturbations). *Suppose that both the true velocity field and the perturbation satisfy the periodic, mean-zero eddy flow assumption. Then, under mild regularity conditions, the NUTS prediction $\hat{\varphi}$ differs from the true solution $\varphi^*$ by at most $\mathcal{O}(\varepsilon)$, independent of the perturbation strength $\|\boldsymbol{\delta}\|_\infty$.*

**Remark:** The formal statement and proof of this result are provided in Appendix C.

**Interpretation.** This theorem establishes two key properties of NUTS. First, the error is $\mathcal{O}(\varepsilon)$ and independent of the perturbation strength $\|\boldsymbol{\delta}\|_\infty$, ensuring robustness: fast, high-amplitude eddy perturbations have negligible impact on the coarse-scale reconstruction. Second, the result guarantees accuracy when the true eddy field varies on small spatial and temporal scales ($\varepsilon \ll 1$). This is nontrivial, as the eddy field enters the dynamics with magnitude $1/\varepsilon$; despite being mean-zero, its local influence is large. The bound confirms that the homogenized model captures the correct large-scale behavior, justifying the use of coarse dynamics in this regime.

## 4 Experiment

In this section, we answer the following research questions:

**RQ1.** How does NUTS perform in reconstructing global surface ocean nutrient concentrations compared to existing baselines, using both simulated and real-world observations?

**RQ2.** Does the proposed two-scale modeling framework enhance robustness to mesoscale perturbations? How do filtering strategies and diffusion implementations influence this robustness?

**RQ3.** How do individual design choices—such as model architecture, auxiliary inputs, conservation loss, and reconstruction interval—affect reconstruction accuracy?

### 4.1 Experimental Setup

We present the experimental setup, including datasets, baselines and evaluation metrics. Implementation and training details are in Appendix D. Code and data are available at URL.

**Data.** We conduct experiments using two datasets for global surface nutrient reconstruction. **Simulation Dataset.** To support high-quality long-term reconstruction, we release two data products generated by the numerical physical-biogeochemical model MOM6-COBALT2 Liu et al. [2022], referred to as MOM6 (Daily) and MOM6 (Monthly). The simulations were conducted on 1000 CPU cores of AMD EPYC 9654 96-Core Processors

Table 1: NRMSEs ($\downarrow$) of MOM6 (Monthly) and GOBH (Monthly) data compared to real observations from WOD.

| Data Source | Nitrate | Phosphate |
|---|---|---|
| MOM6 | 0.463 | 0.301 |
| GOBH | 1.444 | 1.335 |

over an 11-day period, spanning 1959 to 2022 at a global nominal resolution of $0.5°$ ($576 \times 720$). The model output is subsequently regridded to a uniform $0.5°$ grid ($360 \times 720$) using bilinear interpolation. Each data product includes surface nitrate and phosphate concentrations, along with auxiliary variables such as temperature, salinity, and horizontal velocities $(u, v)$. Compared to GOBH (Monthly) Perruche [2018], our MOM6 (Monthly) data product show improved agreement with in-situ observations from WOD, achieving approximately 60% lower NRMSE on a $0.5° \times 0.5°$ grid (Table 1). Additional details are provided in Appendix D.1. **Real Observations.** We use in-situ nutrient measurements from the World Ocean Database (WOD) Mishonov et al. [2024], which contains nitrate and phosphate records from 1959 to 2022. These observations are extremely sparse, covering only **0.16%** of the full spatio-temporal grid. All measurements are regridded to match the spatial and temporal resolution of the MOM6 data product.

**Tasks.** We evaluate the model on two resolution-specific nutrient reconstruction tasks. **Daily Average Reconstruction.** Sparse observations are simulated by randomly sampling nutrient values from the MOM6 (Daily) dataset at sparsity levels of 0.1%, 1%, and 10%. The

Table 2: Overview of dataset divisions by year.

| Task | Train | Validation | Test |
|---|---|---|---|
| Daily Avg. | 2019, 2020 | 2021 | 2022 |
| Monthly Avg. | 1959–1998 | 1999–2010 | 2011–2022 |

0.1% level reflects the sparsity of real-world observations, while 10% aligns with settings used in prior work Luo et al. [2024]. The model reconstructs full daily nitrate and phosphate fields using these samples together with MOM6 daily flow and auxiliary variables. **Monthly Average Reconstruction.** Real-world nutrient measurements from WOD and monthly flow and auxiliary variables from MOM6 are used to reconstruct complete monthly averages of nutrient fields. Dataset partitions are summarized in Table 2, with sampling details in Appendix D.4.

Table 3: NRMSE ($\downarrow$) of different models for reconstructing (1) global daily average nutrient concentrations from the MOM6 simulation under sampling ratios of 0.1%, 1%, and 10%, and (2) global monthly average concentrations from WOD observations. *Params* denotes the number of model parameters. The numbers after $\pm$ are standard errors under 3 trials.

| Methods | Params | MOM6 (Daily) | | | | | | WOD (Monthy) | |
| --- | --- | --- | --- | --- | --- | --- | --- | --- | --- |
| | | Phosphate | | | Nitrate | | | Phosphate | Nitrate |
| | | 0.1% | 1% | 10% | 0.1% | 1% | 10% | – | – |
| Kriging(Exp.) | – | $0.535_{\pm0.022}$ | $0.262_{\pm0.015}$ | $0.184_{\pm0.023}$ | $0.642_{\pm0.020}$ | $0.368_{\pm0.025}$ | $0.256_{\pm0.019}$ | $1.275_{\pm0.130}$ | $1.495_{\pm0.091}$ |
| Kriging(Sph.) | – | $0.537_{\pm0.019}$ | $0.276_{\pm0.022}$ | $0.192_{\pm0.020}$ | $0.649_{\pm0.017}$ | $0.399_{\pm0.018}$ | $0.272_{\pm0.021}$ | $1.270_{\pm0.086}$ | $1.517_{\pm0.057}$ |
| 4D-VarNet | 0.3M | $0.151_{\pm0.008}$ | $0.154_{\pm0.012}$ | $0.156_{\pm0.010}$ | $0.168_{\pm0.006}$ | $0.170_{\pm0.007}$ | $0.161_{\pm0.008}$ | $0.187_{\pm0.008}$ | $0.203_{\pm0.009}$ |
| Marble | 0.6M | $0.397_{\pm0.051}$ | $0.227_{\pm0.044}$ | $0.232_{\pm0.069}$ | $0.441_{\pm0.078}$ | $0.222_{\pm0.044}$ | $0.297_{\pm0.047}$ | $0.363_{\pm0.058}$ | $0.326_{\pm0.056}$ |
| FNO | 4.8M | $0.251_{\pm0.015}$ | $0.227_{\pm0.016}$ | $0.229_{\pm0.014}$ | $0.261_{\pm0.012}$ | $0.256_{\pm0.013}$ | $0.257_{\pm0.014}$ | $0.244_{\pm0.015}$ | $0.276_{\pm0.017}$ |
| U-Net | 31.0M | $0.151_{\pm0.008}$ | $0.148_{\pm0.013}$ | $0.149_{\pm0.011}$ | $0.169_{\pm0.007}$ | $0.166_{\pm0.012}$ | $0.167_{\pm0.013}$ | $0.174_{\pm0.012}$ | $0.187_{\pm0.008}$ |
| ViT | 77.7M | $0.257_{\pm0.032}$ | $0.242_{\pm0.044}$ | $0.359_{\pm0.048}$ | $0.311_{\pm0.046}$ | $0.256_{\pm0.044}$ | $0.256_{\pm0.052}$ | $0.263_{\pm0.034}$ | $0.260_{\pm0.002}$ |
| AtmoRep | 0.7B | $0.196_{\pm0.010}$ | $0.194_{\pm0.011}$ | $0.192_{\pm0.010}$ | $0.190_{\pm0.009}$ | $0.219_{\pm0.011}$ | $0.218_{\pm0.013}$ | $0.206_{\pm0.013}$ | $0.260_{\pm0.013}$ |
| Prithvi | 2.3B | $0.216_{\pm0.055}$ | $0.197_{\pm0.043}$ | $0.208_{\pm0.054}$ | $0.279_{\pm0.049}$ | $0.274_{\pm0.057}$ | $0.275_{\pm0.036}$ | $0.222_{\pm0.042}$ | $0.338_{\pm0.046}$ |
| **NUTS** | 125.6M | $0.014_{\pm0.002}$ | $0.015_{\pm0.001}$ | $0.022_{\pm0.002}$ | $0.143_{\pm0.003}$ | $0.136_{\pm0.003}$ | $0.142_{\pm0.004}$ | $0.035_{\pm0.002}$ | $0.151_{\pm0.003}$ |
| *Promotion* | – | **90.7%** | **89.9%** | **85.2%** | **14.9%** | **18.1%** | **11.8%** | **79.9%** | **19.3%** |

**Baselines.** We compare our model against a wide range of baselines grouped into six categories: (1) Kriging interpolation with exponential and spherical variogram models; (2) CNN-based model U-Net Ronneberger et al. [2015]; (3) transformer-based model ViT Dosovitskiy et al. [2020]; (4) neural operator Fourier Neural Operator (FNO) Li et al. [2020]; (5) implicit representation method Marble model Wang et al. [2025]; (6) foundation models pretrained on climate data, including Prithvi WxC Schmude et al. [2024] and AtmoRep Lessig et al. [2023]; (7) physics-guided hybrid assimilation model such as 4DVarNet Beauchamp et al. [2023]. All baselines except Marble and Kriging reconstruct each frame independently using static inputs—observations, auxiliary variables, and velocity fields at a single time step. Marble leverages temporal observations but excludes auxiliary variables and flow inputs. Kriging uses only static observations. In contrast, our model takes temporal sequences of all inputs and generates spatiotemporal nutrient reconstructions. See Appendix B.2 and D.5 for details.

**Metrics.** We use Normalized Root Mean Squared Error (NRMSE) to evaluate model performance, which ensures scale independence Shcherbakov et al. [2013]. We first calculate the latitude-weighted RMSE between the reconstructed values and the corresponding ground-truth, while NRMSE is obtained by normalizing RMSE using the mean of the ground-truth.

## 4.2 Main Results (RQ1)

We compare the reconstruction performance of our model on simulation and observation data as summarized in Table 3 and Figure 4.

**Obs 1: NUTS achieves the lowest NRMSE across all daily and monthly reconstruction tasks.** We evaluate performance under varying observation sparsity across simulated and real-world datasets. As shown in Table 3, NUTS consistently outperforms all baselines. On the daily task with 0.1% sparsity, it reduces NRMSE by 90.7% for phosphate and 14.9% for nitrate compared to U-Net. On the monthly WOD dataset, it achieves 79.9% and 19.3% improvement, respectively. The gain is more substantial for phosphate, which exhibits smoother temporal variation and is easier to model dynamically. Figure 3 supports this, showing the spatial distribution of the phosphate-to-nitrate ratio of coefficients of variation (CVs), where

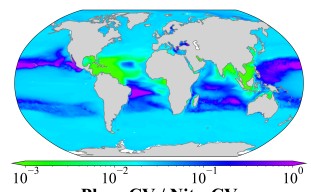

**Phos. CV / Nitr. CV**

Figure 3: Spatial distribution of the phosphate-to-nitrate ratio of coefficients of variation.

each CV is defined as the temporal standard deviation divided by the mean concentration. Lower values indicate weaker phosphate fluctuations, which NUTS captures more reliably.

Among baselines, U-Net and 4D-VarNet perform best. U-Net extracts multiscale features via skip-connected encoders Ronneberger et al. [2015], while 4D-VarNet enforces physical consistency through advection-aware design Beauchamp et al. [2023]. NUTS combines both principles—multiscale modeling and physics-based dynamics—yielding consistent improvements across sparsity levels. These gains are especially pronounced under low observation density, where auxiliary physical variables become essential for accurate reconstruction. Baselines that lack such inputs—such as Kriging (Exp.) and (Sph.)—exhibit large accuracy drops. In contrast, NUTS remains robust by

Table 4: **Model Analysis (NRMSE ↓).** (a) Comparison of different low-pass filter types; (b) Evaluation of cutoff ratios for frequency filtering; (c) Comparison of advection–diffusion implementations, including advection-only, fixed diffusion matrix, and learned diffusion network. Unless otherwise specified, the target nutrient is nitrate, the low-pass filter is Fourier-based with a cutoff ratio of 0.1, and the diffusion module is implemented using a 6-layer ResNet.

(a) **Filter Type.**

| filter | Daily | Monthly |
|---|---|---|
| Fourier | **0.136** | **0.151** |
| Wavelet | 0.144 | 0.197 |
| Gaussian | 0.143 | 0.169 |
| Moving Avg. | 0.154 | 0.167 |

(b) **Filter Cutoff Ratio.**

| param. | Daily | Monthly |
|---|---|---|
| 0.1 | **0.136** | **0.151** |
| 0.2 | 0.145 | 0.194 |
| 0.5 | 0.145 | 0.189 |
| 1.0 | 0.171 | 0.189 |

(c) **Implementation of Advection Diffusion.**

| case | Daily | Monthly |
|---|---|---|
| advection-only | 0.138 | 0.176 |
| adv. + diffusion matrix | 0.142 | 0.165 |
| adv. + diffusion network | **0.136** | **0.151** |

leveraging oceanographic drivers like sea surface temperature, as further confirmed in our ablation study in Section 5.

**Obs 2: In reconstructing real observation site records, our model outperforms data-driven baselines and matches the performance of traditional numerical methods.** We evaluate the site-wise reconstruction accuracy by training on 75% of WOD sites and testing on the remaining 25%. As shown in Figure 4, NUTS achieves site-wise NRMSEs of 1.32 for phosphate and 2.18 for nitrate, outperforming all data-driven baselines. Its performance is comparable to traditional numerical models, including MOM6 and GOBH, demonstrating strong generalization under real-world sparsity.

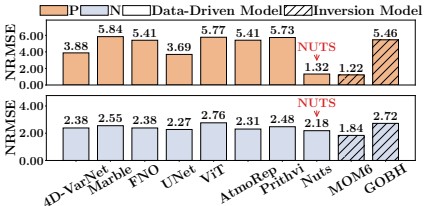

Figure 4: Site-wise NRMSE (↓) of different methods evaluated on WOD real observation.

### 4.3 Component Analysis: Contribution of the Coarse and Refinement Modules (RQ2)

We evaluate the contribution of two key coarse-stage components—low-pass filtering and effective diffusion—as well as the refinement module. NUTS is compared against U-Net and three ablated variants, each omitting a specific component while keeping all other settings fixed. The structural details of these variants are summarized in Table 5, and all variants are parameter-matched with NUTS for a fair comparison. All ablation results reported in this section use nitrate as the reconstruction target. Results for the daily task are reported under a 1% sparsity ratio. Full hyperparameter configurations are provided in Appendix D.5.

**Obs 3: Our model achieves both robustness and accuracy; filtering alone improves stability but sacrifices mesoscale information.** We assess robustness by perturbing the eddy component $\mathbf{v}^*$ using Fourier-based scaling, generating $\boldsymbol{\delta} = \gamma \mathbf{v}^*$, and injecting it into the velocity field. As shown in Figure 5, NUTS maintains low NRMSE across all perturbation levels, demonstrating strong resilience to mesoscale variability. *Naive-B*, which directly propagates the full velocity field without filtering, suffers large errors—especially at $\gamma = \pm 1$—highlighting its sensitivity to unresolved eddy perturbations. *Naive-F* improves

Table 5: Overview of Model Ablation Variants. "B", "F" and "F+D" represent the base model, the base model with filtering, and the base model with both filtering and diffusion, respectively. ✓ denotes inclusion; × denotes exclusion.

| Variants | Params Count | Low-pass Filter | Effective Diffusion | Refine Module |
|---|---|---|---|---|
| *Naive-B* | 131.9M | × | × | × |
| *Naive-F* | 131.9M | ✓ | × | × |
| *Naive-(F+D)* | 131.7M | ✓ | ✓ | × |
| NUTS | 125.6M | ✓ | ✓ | ✓ |

robustness by suppressing high-frequency noise but exhibits degraded accuracy due to the removal of informative mesoscale signals. In contrast, NUTS combines the strengths of both: the coarse stage stabilizes dynamics through filtering, while the refinement stage recovers fine-scale nutrient structure conditioned on residual eddy flow.

**Obs 4: Effective diffusion enhances filtered transport, but refinement is essential for recovering mesoscale structure.** As shown in Figure 5, *Naive-(F+D)*—which combines flow filtering with the effective diffusion module—achieves lower RMSE than *Naive-F* and remains robust under mesoscale perturbations. This validates the use of homogenized advection–diffusion dynamics to stabilize transport and retain partial mesoscale effects. However, despite comparable architecture and parameter count, *Naive-(F+D)* still underperforms our full model, highlight-

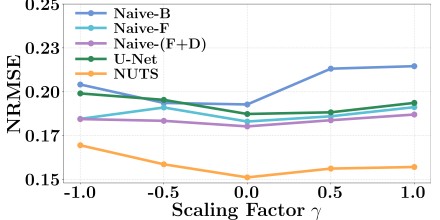

Figure 5: NRMSE of different models under varying mesoscale perturbation levels.

Table 6: **Ablation Study (NRMSE ↓).** **(a)** Comparison of coarse-stage initializers, including static and dynamic architectures; **(b)** Analysis of model depth in the coarse module; **(c)** Analysis of model depth in the refinement module; **(d)** Evaluation of source and conservation loss terms; **(e)** Quantification of the impact of auxiliary input variables; **(f)** Assessment of sensitivity to temporal interval length. All experiments use the default setting: coarse/refine depth of 12/6, all loss terms and inputs included, and interval length set to 4.

(a) **Coarse Model Structure.**

| case | Daily | Monthly |
|------|-------|---------|
| 2D CNN | 0.206 | 0.161 |
| ViT | 0.159 | 0.157 |
| 3D CNN | 0.162 | 0.153 |
| NUTS | **0.136** | **0.151** |

(b) **Depth of Coarse Module.**

| depth | Daily | Monthly |
|-------|-------|---------|
| 6 | 0.185 | 0.160 |
| 8 | 0.165 | 0.153 |
| 12 | **0.136** | **0.151** |
| 16 | 0.148 | 0.177 |

(c) **Depth of Refine Module.**

| depth | Daily | Monthly |
|-------|-------|---------|
| 2 | 0.142 | 0.164 |
| 4 | 0.146 | 0.210 |
| 6 | **0.136** | **0.151** |
| 8 | 0.180 | 0.170 |

(d) **Source and Conservation Loss.**

| case | Daily | Monthly |
|------|-------|---------|
| w/ src, w/ cons. | **0.136** | **0.151** |
| w/ src, w/o cons. | 0.153 | **0.151** |
| w/o src, w/ cons. | 0.156 | 0.155 |
| w/o src, w/o cons. | 0.155 | 0.154 |

(e) **Auxiliary Variables.**

| removed var. | Daily | Monthly |
|--------------|-------|---------|
| temp. | 1.059 | 1.014 |
| salt | 0.170 | 0.168 |
| u | 0.164 | 0.155 |
| v | 0.142 | 0.160 |

(f) **Interval Length.**

| length | Daily | Monthly |
|--------|-------|---------|
| 1 | 0.159 | 0.157 |
| 2 | 0.166 | **0.151** |
| 4 | **0.136** | **0.151** |
| 8 | 0.220 | 0.158 |

ing the importance of the refinement module in reconstructing fine-scale nutrient variability lost during filtering.

**Obs 5: Filter design in the coarse module is critical; Fourier filtering with strong high-frequency suppression yields the best performance.** We ablate the design of the low-pass filter used in the coarse module of NUTS. Among several options, the Fourier filter achieves the lowest NRMSE on both daily (0.136) and monthly (0.151) tasks, outperforming wavelet, Gaussian, and moving average filters (Table 4a). This result is consistent with prior work in ocean modeling and geophysical fluid dynamics Abernathey and Marshall [2013], Callies and Ferrari [2013], where spectral (Fourier-based) filtering is widely adopted to separate large-scale flow from unresolved mesoscale variability. We further vary the cutoff ratio of the Fourier filter, which determines the extent of high-frequency suppression. Lower ratios—removing more unresolved eddy components—consistently improve reconstruction accuracy, while higher ratios degrade performance (Table 4b). These results highlight that principled filtering in the coarse module is essential for stabilizing nutrient transport, while fine-scale variability is later recovered by the refinement stage.

**Obs 6: Incorporating a learnable diffusion module improves accuracy; state-dependent designs further enhance performance.** We ablate the diffusion design in the advection–diffusion solver of NUTS. We compare three variants: (1) advection-only, (2) with a trainable, time-invariant diffusion matrix $K = UU^\top$, and (3) the state-dependent formulation used in **NUTS**, where $K = GG^\top$ and $G = G(\bar{\varphi})$ is produced by a hyper-network conditioned on the coarse prediction $\bar{\varphi}$. As shown in Table 4c, both diffusion-enhanced variants outperform the advection-only baseline on daily and monthly tasks, confirming the benefit of modeling unresolved subgrid dispersion. The state-dependent design used in NUTS further improves accuracy over the time-invariant variant (0.151 vs. 0.165 on the monthly task), consistent with the theoretical expectation that effective diffusion depends on the tracer state McWilliams [2006], McDougall and McIntosh [2001]. The improvement is more substantial in the monthly setting, where longer temporal scales allow diffusion to play a more dominant role in shaping nutrient transport.

## 5 Discussions

**Ablation Study (RQ3).** We evaluate the impact of architectural selection of both modules, source module, loss design, auxiliary variables and temporal interval length on model performance. Additional ablation results on spatial and temporal resolution, as well as the conservation loss weight coefficient, are provided in Appendix E.1. All ablation results in this section use nitrate as the target variable. ● **Coarse and Refine Module Structure.** We evaluate architecture and depth for both the coarse and refinement modules. As shown in Table 6a, static 2D CNNs underperform due to the lack of temporal modeling, while dynamic architectures—3D CNN and spatiotemporal ViT—achieve lower errors. NUTS, which uses a spatiotemporal transformer, yields the best NRMSE of 0.151. Depth analysis (Tables 6b, 6c) shows that performance peaks with 12 layers in the coarse module and 6 layers in the refinement module. Shallower models underfit, while deeper ones degrade due to over-smoothing or training instability. These results highlight the importance of both dynamic structure and moderate depth. ●**Source and Conservation Loss.** Incorporating the source module

and conservation loss enhances reconstruction accuracy (Table 6d). Additionally, the conservation loss contributes to preserving total nutrient mass, as detailed in Appendix E.2. ●**Auxiliary Variables.** Sea surface temperature is the most influential auxiliary input, with its removal causing the largest increase in reconstruction error (Table 6e). This highlights its essential role in guiding nutrient reconstruction and is consistent with prior findings in related work such as 4DVarNet Beauchamp et al. [2023]. ●**Reconstruction Interval Length.** Model performance is sensitive to the choice of reconstruction interval length, with both short and long intervals resulting in higher error relative to intermediate settings (Table 6f). In the daily task, a 4-step interval yields the lowest NRMSE (0.136), balancing informative temporal context and noise from redundancy or uncorrelated variability.

**Conclusion and Broader Impact.** We present NUTS, a two-stage, physics-informed framework for reconstructing global surface ocean nutrients from sparse observations. By combining coarse advection–diffusion dynamics with data-driven refinement, NUTS achieves state-of-the-art performance on both simulated and real-world datasets. While our experiments focus on nitrate and phosphate, the framework is grounded in general transport physics and naturally extends to other passive tracers. Preliminary results (Appendix F) show promising generalization, supporting broader applications in environmental reconstruction, climate monitoring, and Earth system science.

**Future Work.** Future directions include extending NUTS in three areas: spatial coverage, biogeochemical complexity, and air–sea exchange. A 3D extension will capture vertical transport and subsurface gradients. Adding processes like remineralization and nutrient uptake will improve modeling of regeneration and biological consumption. Air–sea gas exchange will enable reconstruction of gas tracers for carbon and oxygen cycle monitoring.

## Acknowledgements

This research is supported by the National Natural Science Foundation of China (No. 62306179), the National Key Research and Development Program of China (2023YFC2808802), Southern Marine Science and Engineering Guangdong Laboratory (Zhuhai) (nos. SML2023SP219), the Ocean Negative Carbon Emissions (ONCE) Program.

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
