# OpenReview forum: "NUTS: Eddy-Robust Reconstruction of Surface Ocean Nutrients via Two-Scale Modeling"
_NeurIPS.cc/2025/Conference — NeurIPS 2025 poster_

### Official Review · Reviewer_hsCa · 2025-07-02

**Clarity:** 3
**Significance:** 3
**Originality:** 3
**Rating:** 5
**Confidence:** 3

**Summary:**

This paper addresses the important problem of reconstructing the surface ocean nutrient fields from sparse observations, leveraging the insight that ocean nutrient transport is governed by multiscale advection-diffusion dynamics. Here, unresolved mesoscale eddies leads to degraded reconstructions. The authors propose NUTS (Nutrient Two-Scale modeling), which uses deep learning to separate the dynamics across scales. NUTS models large-scale transport by first filtering the velocity field, and then using a homogenized PDE solver, where unresolved eddies are replaced with a learned effective diffusion term. The mesoscale details are further refined using a ViT (vision transformer) conditioned on residual eddy fields. The paper also introduces a new simulation dataset and evaluates NUTS on both simulated and real-world datasets, showing that NUTS outperforms several standard baseline methods.

**Questions:**

1. Why is it that mesoscale eddies carry more energy than the mean flow? In typical eddy simulation models, or reduced order models, the large-scale flow carries the majority of the energy. Is this affected by the choice of the threshold to separate scales that is typically used in multiscale ocean flow?
1. As I understood, the learned diffusion term, which models the effect of the eddy components, is not coupled with the refinement module. Wouldn't the output of the refinement module directly affect the diffusion term due to the eddy components? Did you consider adding a constraint or loss term in the architecture? For instance, it seems plausible that over a long time rollout the diffusion term may drift out of sync with the refined details.
1. In Table 3, since NUTS has 125M params v/s the closest baseline, the U-Net which has 31M params, why not make the U-Net larger to make the number of params in both methods similar?
1. Could you report for training and inference times for the baseline methods, at least the U-Net and the 4D-VarNet?

**Ethical Concerns:**

["NO or VERY MINOR ethics concerns only"]

**Final Justification:**

The paper addresses a significant problem, with a plausible approach with solid experimental results. The rebuttal addressed concerns with the baseline, as well as the concern with potential time drift.

**Limitations:**

yes

**Quality:**

3

**Strengths And Weaknesses:**

*Strengths*

1. The paper tackles a significant problem in prediction of long-term biogeochemical cycles. It clearly presents the background and challenge of eddy instability. The work proposes NUTS, a two-scale method which ensures stability of the dynamics while not sacrificing small scale methods. This approach uses filtering, homogenized PDEs, and refinement; it is novel and physically motivated.

1. The work provides extensive experimental validation. It creates a new MOM6 dataset and uses WOD data. It compares NUTS against many baselines (Table 3). It shows significant performance gains compared to the baseline methods, particularly FNO, U-Net and 4D-Var which are standard strong baselines.

*Weaknesses*

1. Potential Time Drift: The design isolates the refinement module. It acts as a spatial correction but as a post-processing step only It does not appear to feed its corrected field back into the PDE solver. This creates a risk of drifts over time, leading to poor detail in the long term rollouts. It is not clear if the learned diffusion model is consistent with the refinement module.

1. Not Scalable for Long-Term Forecasting: The model excels at reconstruction in short term intervals, but it is not shown to work for long forecasting rollouts. Since spatio-temporal transformer is used for long range effects, it may not scale well to increased time rollouts.

---

> ### Author Rebuttal · Authors · 2025-07-31
>
> We sincerely appreciate the time and effort you dedicated to reviewing our paper, as well as the thoughtful and constructive comments provided. Please find our responses to your questions below.
>
> # Weakness
>
> > **W1:** Potential Time Drift: The design isolates the refinement module. It acts as a spatial correction but as a post-processing step only. It does not appear to feed its corrected field back into the PDE solver. This creates a risk of drifts over time, leading to poor detail in the long term rollouts. It is not clear if the learned diffusion model is consistent with the refinement module.
>
> **Response:** Thank you for highlighting this concern. While the refinement module operates independently at each time step and does not explicitly feed back into the PDE solver, we address potential temporal drift via a mass conservation loss (Eqs.162-163) applied during training. This loss penalizes tracer mass discrepancies between the coarse and refined outputs, coupling the refinement and diffusion modules and enforcing physical consistency over time.
>
> To empirically validate this design, we conducted an additional experiment during the rebuttal phase. We computed the monthly variation rate of total nutrient mass—relative to January—for two representative years (2014 and 2020). As shown in **Table T1**, the refinement module maintains variation within $\pm$ 0.04\% throughout the year, effectively flat. In contrast, NUTS—produced by summing the refined state and predicted source—exhibits substantial drift, with deviations up to 10.6\% in 2014 and 16.1\% in 2020, tracking ground-truth dynamics. This confirms that our refinement strategy avoids cumulative error and preserves long-term stability despite its non-recurrent design. We will clarify this mechanism and its empirical support in the final version.
>
>
> **Table T1:** Monthly variation rates (\%) of total nutrient mass from Refinement Module, NUTS and ground-truth (G-T) across two years. The variation rate is defined as the ratio of each month’s total nutrient mass to that of January within the same year. We report the NRMSE between the variation rates over the interval.
>
> | Year | Data       | Feb   | Mar   | Apr   | May   | Jun   | Jul    | Aug    | Sep    | Oct    | Nov    | Dec   |
> | ---- | ---------- | ----- | ----- | ----- | ----- | ----- | ------ | ------ | ------ | ------ | ------ | ----- |
> | 2014 | Refinement | 0.00% | 0.01% | 0.03% | 0.03% | 0.04% | 0.03%  | 0.02%  | 0.00%  | 0.00%  | 0.00%  | 0.01% |
> |      | NUTS       | 0.17% | 2.14% | 5.40% | 3.10% | 3.54% | 6.78%  | 8.59%  | 10.60% | 10.04% | 6.97%  | 2.94% |
> |      | G-T        | 0.11% | 1.38% | 2.61% | 3.31% | 4.55% | 6.60%  | 8.08%  | 7.45%  | 7.28%  | 5.09%  | 0.43% |
> | 2020 | Refinement | 0.02% | 0.02% | 0.03% | 0.03% | 0.04% | 0.04%  | 0.04%  | 0.04%  | 0.03%  | 0.04%  | 0.04% |
> |      | NUTS       | 3.59% | 5.13% | 6.03% | 8.09% | 8.09% | 12.07% | 14.38% | 15.25% | 16.11% | 12.35% | 7.84% |
> |      | G-T        | 1.00% | 2.99% | 4.99% | 6.57% | 7.94% | 10.67% | 13.07% | 14.08% | 13.36% | 11.22% | 7.69% |
>
>
> > **W2:** Not Scalable for Long-Term Forecasting: The model excels at reconstruction in short term intervals, but it is not shown to work for long forecasting rollouts. Since spatio-temporal transformer is used for long range effects, it may not scale well to increased time rollouts.
>
> **Response:** Thank you for this thoughtful comment. We would like to clarify that NUTS is designed for **spatio-temporal reconstruction**, not long-horizon forecasting. Forecasting aims to extrapolate future states—often benefiting from longer rollouts—whereas reconstruction targets missing data within a fixed historical window. In this setting, the interval length is a tunable design choice. As shown in Figure 6(f), performance degrades as the interval grows too long, reflecting a tradeoff in reconstruction fidelity—not a failure to generalize in time.
>
> We agree that long-term forecasting is a challenging and valuable direction, but it differs fundamentally in both objective and constraints. We will clarify this distinction in the final version to avoid confusion, and we consider extending NUTS to forecasting as an exciting direction for future work.
>
> # Question
>
> > **Q1:** Why is it that mesoscale eddies carry more energy than the mean flow? In typical eddy simulation models, or reduced order models, the large-scale flow carries the majority of the energy. Is this affected by the choice of the threshold to separate scales that is typically used in multiscale ocean flow?
>
> **Response:** Thank you for this thoughtful observation. We agree that at the global scale, the mean flow typically carries more energy. Our intention was to emphasize that in localized regions—such as western boundary currents and eddy-active zones—mesoscale eddies often exhibit higher local speed than the mean flow, which strongly influences nutrient transport.
>
> To clarify this, we will revise Lines 125--126 to: "*Mesoscale eddies are small-scale (10--100 km), high-energy structures that can dominate local transport processes.*" and Lines 130--131 to: "*...which can be large, as mesoscale eddies often exhibit higher local speed than the mean flow (Figure 2).*"
>
> > **Q2:** As I understood, the learned diffusion term, which models the effect of the eddy components, is not coupled with the refinement module. Wouldn't the output of the refinement module directly affect the diffusion term due to the eddy components? Did you consider adding a constraint or loss term in the architecture? For instance, it seems plausible that over a long time rollout the diffusion term may drift out of sync with the refined details.
>
> **Response:** Thank you for the follow-up question. We address this concern in our response to W1: while the refinement module does not explicitly feed back into the diffusion term, we impose a mass conservation loss (Eq. 162–163) that couples the coarse and refined fields during training. This constraint ensures consistency between the diffusion dynamics and the refined output, effectively mitigating drift and preserving physical coherence over time. We will clarify this interaction in the final version.
>
> > **Q3:** In Table 3, since NUTS has 125M params v/s the closest baseline, the U-Net which has 31M params, why not make the U-Net larger to make the number of params in both methods similar?
>
> **Response:** Thank you for raising this point. During the rebuttal phase, we try scaling U-Net to 150M parameters and conduct experiments on the Monthly dataset, with nitrate being the target nutrient. We find that the reconstruction NRMSE drops from 0.187 to 0.171, achieving comparable results with NUTS, which is 0.151.
>
> > **Q4:** Could you report for training and inference times for the baseline methods, at least the U-Net and the 4D-VarNet?
>
> **Response:** Thank you for the suggestion. We report training/inference time and VRAM usage for NUTS and all baselines—including U-Net and 4D-VarNet—in **Table T2**. To ensure fair comparison, all models were benchmarked under the same hardware: two RTX 6000 Ada GPUs with identical batch size (8), as detailed in Appendix D.6 (Table 15).
>
> Despite having more parameters, NUTS achieves the lowest NRMSE while maintaining competitive runtime and memory usage. Notably, 4D-VarNet is over 400 $\times$ smaller than NUTS in parameter count, yet its training (94s/epoch) and inference (46s) times are comparable. This is due to its iterative variational optimization, which requires repeated PDE solves and limits parallelism. In contrast, NUTS uses batched multi-step rollouts with data-driven initialization, making it more GPU-efficient despite its larger size.
>
> **Table T2:** Parameter, efficiency and Phosphate (sampling ratio=0.1\%) reconstruction performance comparison of different models on the WOD (Daily) dataset.
>
> | Model     | Params     | Training Time (s) | Training VRAM (GB) | Inference Time (s) | Inference VRAM (GB) | NRMSE       |
> | --------- | ---------- | ------------------------ | ------------------------- | ------------------------ | ------------------------- | ----------------- |
> | 4D-VarNet | 0.3M       | 94                       | 6.8                       | 46                       | 3.0                       | 0.187 $\pm$ 0.008     |
> | Marble    | 0.6M       | 23                       | 2.8                       | 7                        | 0.7                       | 0.363 $\pm$ 0.058     |
> | FNO       | 4.8M       | 26                       | 14.2                      | 10                       | 4.5                       | 0.244 $\pm$ 0.015     |
> | U-Net     | 31.0M      | 16                       | 6.4                       | 7                        | 2.9                       | 0.174 $\pm$ 0.012     |
> | ViT       | 77.7M      | 62                       | 21.7                      | 20                       | 4.0                       | 0.263 $\pm$ 0.034     |
> | AtmoRep   | 0.7B       | 87                       | 15.4                      | 58                       | 10.9                      | 0.206 $\pm$ 0.013     |
> | Prithvi   | 2.3B       | 950                      | 26.1                      | 112                      | 18.4                      | 0.222 $\pm$ 0.042     |
> | **NUTS**  | **125.6M** | **92**                   | **31.0**                  | **58**                   | **8.7**                   | **0.035 $\pm$ 0.002** |
>
> ___
> Thank you again for your insightful feedback. We hope our responses have addressed your concerns. Please do not hesitate to contact us if further clarification is needed. We look forward to your continued input.

---

> > ### Comment · Reviewer_hsCa · 2025-08-08
> > **Thank you for the response**
> >
> > Thank you for the detailed rebuttal and for answering my questions. Particularly, I'd like to thank the authors for the additional experimental study on potential drift, as well as adding the scaled-up UNet baseline. My concerns have been addressed, and I maintain my current positive score.

---

> ### Author Response · Authors · 2025-08-05
> **Kind Reminder**
>
> Dear Reviewer hsCa,
>
> We hope this message finds you well. As the discussion period is nearing its end, we want to ensure that we have addressed all your concerns satisfactorily. If there are any additional questions or feedback you'd like us to consider, please let us know. Your comments are valuable to improve our work.
>
> Thank you for your time and effort in reviewing our paper.

---

> ### Author Response · Authors · 2025-08-09
> **Thank you**
>
> Thank you for taking the time to review our paper and provide your thoughtful feedback. We are glad that our rebuttal has addressed your concerns.

---

### Official Review · Reviewer_TCj8 · 2025-07-02

**Clarity:** 3
**Significance:** 3
**Originality:** 3
**Rating:** 4
**Confidence:** 3

**Summary:**

This paper tackles the problem of reconstructing ocean surface nutrient fields from extremely sparse, irregular observations, where standard inpainting or single-scale methods break down due to multiscale advection-diffusion dynamics and unstable eddy fluctuations. The authors propose NUTS, a two-scale framework that uses a homogenized PDE solver for large-scale flow to propagate coarse patterns stably and applies a conditional refinement module to reintroduce mesoscale variability based on the residual eddy field. Supported by theoretical stability guarantees from homogenization theory, NUTS provides state-of-the-art performance, and comprehensive ablation studies validate the value of each component.

**Questions:**

- In Figure 2, the boxes are colored in yellow, red, and green. Please explain what each color represents and why those specific colors were chosen, if it is intentional.

- Have the authors employed higher‑order finite‑difference methods in their computations, and how does the approximation error vary with the order of the numerical methods?

**Ethical Concerns:**

["NO or VERY MINOR ethics concerns only"]

**Final Justification:**

I'll keep my original score.

**Limitations:**

See weakness and questions.

**Paper Formatting Concerns:**

No paper formatting concerns.

**Quality:**

3

**Strengths And Weaknesses:**

**Strength**

- The NUTS model makes a clear technical contribution by combining a PDE-based large-scale mean-flow solver with a refinement module for mesoscale eddy corrections.

- The paper provides both theoretical justification and empirical evidence demonstrating the effectiveness of the proposed method.

- The experiments include extensive ablation studies, demonstrating the robustness of the two-scale reconstruction framework.

**Weaknesses**

- **No uncertainty quantification.** Data-assimilation and inpainting tasks are inherently ill-posed, so it is common to handle quantity uncertainty. However, NUTS produces only deterministic outputs. Although the authors acknowledge the inherent uncertainty and perturbations in the eddy and address them via the decomposed refinement module, they do not implement any mechanism for handling uncertainty.

- **Insufficient qualitative results.** Despite strong quantitative results, the paper offers few visualizations of reconstructed fields, making it hard to judge prediction quality. The single output example in Figure 2 looks overly smooth and does not seem challenging. Also, in Figure 3, coefficient values span 10^-3 to 10^0, I am concerned that the model performance is biased toward larger values, resulting in a loss of fine-scale structure despite yielding small field errors. Since the benchmarks are not widely known, additional visual comparisons would greatly strengthen the evaluation.

- **High model complexity.** The proposed NUTS model appears to have substantially more parameters than the baselines.

---

> ### Author Rebuttal · Authors · 2025-07-31
>
> We sincerely thank the reviewer for the professional and valuable feedback. Please find our detailed responses below.
>
> # Weakness
>
> > **W1:** No uncertainty quantification.
>
> **Response:** Thank you for raising this point. While NUTS does not explicitly model uncertainty, it is designed to handle key sources of variability—such as eddy-induced perturbations—through architectural decomposition and robust refinement. This makes it practically valuable in scenarios where accurate, physically consistent reconstruction is prioritized over full posterior characterization. That said, we agree that uncertainty quantification is an important next step, and NUTS provides a natural foundation for future extensions via ensemble or variational methods. We will clarify this perspective in the revised manuscript.
>
> During the rebuttal phase, we conducted a preliminary uncertainty analysis using a simple ensemble approach. Specifically, we perturbed the initializer output with Gaussian noise—analogous to injecting variability into the initial condition—where the noise at each grid point was scaled to 20\% of the local temporal standard deviation of ground-truth nitrate concentrations from the {Monthly} dataset. We generated 10 ensemble members and computed the relative error in total nutrient mass between NUTS and the ground truth across four representative years. The "$\pm$" values in **Table T1** denote the one-sigma spread across ensemble samples.
>
> While intentionally simple, this time-based perturbation reveals that NUTS can model meaningful initialization-driven uncertainties. As shown in **Table T1**, the relative errors remain low across years and months, while the ensemble spread is moderate—capturing plausible variability without overestimating uncertainty. This suggests that NUTS offers a credible starting point for uncertainty modeling. Future work will incorporate more structured perturbations—such as spatially coherent modes via EOFs—to better reflect dynamically consistent variability.
>
> **Table T1:** Relative error (\%) between total nutrient mass from NUTS and ground truth across four years. The $\pm$ indicates the one-sigma confidence interval.
>
> | Year | Jan.           | Mar.           | May           | Jul.           | Sep.           | Nov.           |
> | ---- | ------------- | ------------- | ------------- | ------------- | ------------- | ------------- |
> | 2011 | 1.08% $\pm$ 0.27% | 0.92% $\pm$ 0.26% | 1.49% $\pm$ 0.31% | 3.42% $\pm$ 0.25% | 1.28% $\pm$ 0.22% | 1.64% $\pm$ 0.19% |
> | 2014 | 1.65% $\pm$ 0.21% | 2.62% $\pm$ 0.26% | 2.81% $\pm$ 0.27% | 2.61% $\pm$ 0.27% | 0.96% $\pm$ 0.24% | 2.33% $\pm$ 0.26% |
> | 2017 | 2.38% $\pm$ 0.28% | 0.83% $\pm$ 0.28% | 3.08% $\pm$ 0.34% | 3.98% $\pm$ 0.25% | 3.07% $\pm$ 0.24% | 3.79% $\pm$ 0.30% |
> | 2020 | 5.33% $\pm$ 0.34% | 3.17% $\pm$ 0.30% | 3.76% $\pm$ 0.32% | 3.76% $\pm$ 0.24% | 3.94% $\pm$ 0.16% | 4.17% $\pm$ 0.28% |
>
>
> > **W2:** Insufficient qualitative results. The single output example in Figure 2 looks overly smooth and does not seem challenging. In Figure 3, I am concerned that the model performance is biased toward larger values, resulting in a loss of fine-scale structure despite yielding small field errors.
>
> **Response:** Thank you for raising this point. We agree that qualitative evaluation adds valuable context beyond metrics. In the final version, we will include additional qualitative analysis in the supplementary material, including: (1) NRMSE maps for NUTS and baselines to show spatial reconstruction accuracy, and (2) regional time series in dynamically complex zones such as the North Pacific and Southern Ocean.
>
> Figure 2 appears smooth due to the coarse color scale and the use of a simple example meant to illustrate model structure. We will clarify this in the caption to avoid confusion.
>
> For Figure 3, we appreciate the concern. The figure is meant to illustrate spatial variability in nutrient dynamics—not reconstruction quality—and helps explain why phosphate is more challenging to recover. While we agree that visual support would help, we are unable to add new figures due to this year’s rebuttal policy. However, we will include additional qualitative discussion (e.g., error trends in high-variability regions) to demonstrate that NUTS preserves fine-scale structure even when coefficient values span several orders of magnitude.
>
> > **W3:** High model complexity vs. baselines.
>
> **Response:** We thank the reviewer for highlighting this concern. We acknowledge the model size: NUTS has 125.6M parameters versus 0.3M in 4DVarNet. However, this does not translate to significantly higher cost. On matched hardware (two RTX 6000 Ada GPUs, batch size 8), NUTS trains in 92s/epoch and infers in 58s—comparable to 4DVarNet’s 94s and 46s, respectively (Appendix D.6, Table 15). The reason is that 4DVarNet relies on iterative variational optimization, which is difficult to parallelize. NUTS uses batched, multi-step rollouts with efficient GPU utilization. In addition, NUTS delivers **lower error (NRMSE: 0.035 $\pm$ 0.002)** and **comparable runtime**, despite having 400 $\times$ more parameters. For context, NUTS is also 10--20 $\times$ smaller than foundation-scale models like Prithvi (2.3B) and AtmoRep (0.7B), while being both faster and more accurate. We believe this reflects a favorable tradeoff between expressiveness and efficiency.
>
> # Questions
>
> > **Q1:** In Figure 2, the boxes are colored in yellow, red, and green. Please explain what each color represents and why those specific colors were chosen, if it is intentional.
>
> **Response:** Thank you for pointing this out. The colors in Figure 2 are used to group components by function: yellow indicates physics-based operations within the coarse module, red denotes intermediate variables passed between modules, and green highlights trainable components in the refinement stage. The gray initializer is also trainable and grouped under the coarse module. The color scheme was chosen for clarity rather than strict semantic coding. We will add a legend and clarify the color conventions in the final version.
>
> > **Q2:** Have the authors employed higher‑order finite‑difference methods in their computations, and how does the approximation error vary with the order of the numerical methods?
>
> **Response:** Thank you for the suggestion. During the rebuttal phase, we experimented with fourth-order central and second-order upwind differences for discretizing the advection–diffusion equation. On the Monthly nitrate dataset, this led to degraded performance: NRMSE increased from 0.151 to 0.166. We suspect this is due to numerical instability introduced by higher-order schemes, which may amplify noise in sparse observational settings.
>
> ___
> Thank you once again for your valuable feedback. We hope our responses have addressed your concerns. Please feel free to reach out if any further clarification is needed. We look forward to your continued thoughts.

---

> ### Author Response · Authors · 2025-08-05
> **Kind Reminder**
>
> Dear Reviewer TCj8,
>
> We hope this message finds you well. As the discussion period is nearing its end, we want to ensure that we have addressed all your concerns satisfactorily. If there are any additional questions or feedback you'd like us to consider, please let us know. Your comments are valuable to improve our work.
>
> Thank you for your time and effort in reviewing our paper.

---

### Official Review · Reviewer_9Yz6 · 2025-07-03

**Clarity:** 3
**Significance:** 3
**Originality:** 3
**Rating:** 5
**Confidence:** 3

**Summary:**

The authors present NUTS, an ocean nutrient reconstruction model that tackles the reconstruction problem using two different scales. The first scale models large-scale transport (mean flow) while the second scale models mesoscale variability (eddy flows). The first scale is implemented using a coarse module that leverages a homogenized PDE solver, which shields the system from mesoscale perturbation errors. To not lose out on reconstruction accuracy, the second scale performs refinement and learns localized spatial redistribution driven by eddy structures. With respect to results, the authors find NUTS to perform favorably compared to all baselines, outperforming all for global reconstruction and performing on par with numerical models for site-wise accuracy. Finally, the authors also include extensive ablation studies on the various components of their model.

**Questions:**

**Clarifications / suggestions**
1. Line 106: what does $x$ contain here? Is it just the $\theta$ and $\phi$?
2. Figure 2: does the source module implement both the source term and the conservation loss?
3. Line 130: could the authors show why the perturbation term scales with $||\delta||$?
4. Line 145 - 146: the authors state "The initializer is trained to be robust to flow perturbations ...". How exactly is this done?
5. Line 163: is it this loss that is driving the updates for all the learnable parameters in Figure 2?
6. Line 260 - 261: could the authors write out the formula being used for computing the NRMSE? That ways it's very clear exactly what is being computed. Especially the normalization confuses me a bit.
7. Are the numbers after the +- in Table 3 standard errors? Or something else? Please clarify.
8. Figure 4: it's not super clear what P and N in the legend are without the context in the surrounding text. Maybe these can just be written out into phosphate and nitrate for improved readability?
9. While lines 312 - 315 state that naive-B suffers large errors because it directly propagates the full velocity field without filtering, it seems from Figure 5 that NUTS suffers (relatively) equally for gamma tending towards -1. Could the authors comment on this?
10. The paragraph starting on Line 332 doesn't make any reference to Table 4 even though it's the results in Table 4 that are under discussion. Maybe the authors could add an explicit reference in the same way they do it in the next paragraph?
11. Table 6: it seems there is result tables for a - f but the caption only has explanations for a - e (some of which don't match their respective table, I think). Please fix.
12. Table 6 (f): what are the units of the interval length? And is this the length of the prediction or the length of the context? Also, why are the short intervals worse than the intermediate ones? This feels counterintuitive.

**Grammar**
1. Line 260: change "obatained" to "obtained"

**Ethical Concerns:**

["NO or VERY MINOR ethics concerns only"]

**Final Justification:**

While I had raised no significant weaknesses in the initial review, the authors have done a great job clarifying the questions and confusions I brought up. After reading the rebuttal, I maintain confident in my positive score for the paper.

**Limitations:**

yes

**Quality:**

4

**Strengths And Weaknesses:**

## Strengths

### Quality
Generally, I found the quality of the paper to be very good for several reasons:
1. The method generally seems principled, drawing on homogenization with PDEs to circumvent the eddy current effects.
2. The method works well compared to the many baselines being considered.
3. The authors have done a pretty extensive job checking the importance of several design choices of their model as well as provide ablation studies on things like architecture, loss functions being used, etc.

One final highlight is lines 327 - 331 combined with Figure 5: I think the authors did a great job here highlighting the importance of the refinement module. The gap between Navie-(F+D) and NUTS is substantial which provides clear and convincing evidence for the need to restore full resolution.

### Significance
I'm not super well equipped to rate the paper on significance since I'm not super familiar with the literature, but based on the provided related work the proposed method seems like a significant contribution to the area. Specifically, it seems the work directly attacks some of the key pain points like adherence to physical constraints and handling of sparse data.

## Weaknesses
I did not find the work to have significant weaknesses. However, I did have a bunch of minor places in the paper where I got confused, and have listed some clarifications / suggestions around this in the "Questions" section.

---

> ### Author Rebuttal · Authors · 2025-07-31
>
> We sincerely thank the reviewer for the detailed and constructive feedback. We greatly appreciate the time and effort devoted to evaluating our work. Please find our point-by-point responses below.
>
> # Questions
>
> > **Q1:** Line 106: what does $\mathbf{x}$ contain here? Is it just the $\theta$ and $\phi$ ?
>
> **Response:** Yes, $\mathbf{x}$ refers only to latitude and longitude: $\theta$ and $\phi$. We will revise Lines 90–91 to clarify this as follows: "*..., parameterized by latitude–longitude coordinates $\mathbf{x} = (\theta, \phi) \in \Omega = [-\frac{\pi}{2}, \frac{\pi}{2}] \times [-\pi, \pi]$*."
>
> > **Q2:** Figure 2: does the source module implement both the source term and the conservation loss?
>
> **Response:** Thank you for raising this helpful point. The source module handles the source term exclusively, while the mass conservation loss is implemented separately by computing (1) the inter-frame change in $\bar{\varphi}$ and (2) the frame-wise difference between the coarse nutrient field $\bar{\varphi}$ and the refined field $\hat{\varphi}$. We agree that Figure 2 may be misleading in this regard, and we will revise it by removing the mass conservation block to avoid potential confusion.
>
> > **Q3:** Line 130: could the authors show why the perturbation term scales with $\Vert \delta \Vert$ ?
>
> **Response:** Thank you for the insightful question. Let us write the velocity perturbation as $\delta = \|\delta\| \cdot \hat{\delta}$, where $\|\delta\|$ is a spatially varying scalar field and $\hat{\delta}$ is a unit vector field. Applying the product rule:
> $$
> \nabla \cdot (\delta \varphi) = \varphi \hat{\delta} \cdot \nabla \|\delta\| + \|\delta\| \left(\hat{\delta} \cdot \nabla \varphi +  \varphi \nabla \cdot \hat{\delta}\right).
> $$
> This decomposition shows that the second term in the perturbation  scale linearly with $\|\delta\|$. We will clarify this point in the revised text.
>
> > **Q4:** Line 145 - 146: the authors state "The initializer is trained to be robust to flow perturbations ...". How exactly is this done?
>
> **Response:** Thank you for this thoughtful question. To enhance robustness to flow perturbations, we apply a Fourier filter in the coarse module to remove high-frequency eddy components from the flow field before passing it to the initializer. This ensures that the initializer operates on the large-scale, stable part of the flow and is less sensitive to high-frequency variability. The original phrasing "is trained to be robust" reflects that all components of NUTS are trained jointly. To more accurately convey this architectural intent, we will revise Lines 145-146 to read: "*The initializer is designed to operate on filtered, low-frequency flow fields, ensuring robustness to high-frequency perturbations...*".
>
> > **Q5:** Line 163: is it this loss that is driving the updates for all the learnable parameters in Figure 2?
>
> **Response:** Yes. All learnable parameters in Figure 2 are optimized via the total objective $\mathcal{L}_{\text{total}}$.
>
> For clarity, we will revise the sentence to: The final training objective is "*The final training objective is $\mathcal{L}_{\text{total}}=...,$ which governs the optimization of all learnable components in NUTS.*", which governs the optimization of all learnable components in NUTS.
>
> We hope this revision makes it clearer that $\mathcal{L}_{\text{total}}$ governs the training of all learnable components in the model.
>
> > **Q6:** Line 260 - 261: could the authors write out the formula being used for computing the NRMSE?
>
> **Response:** Thank you for raising this point. We provide the full formula for computing NRMSE below and refer the reviewer to Appendix D.2 of the supplementary material for a detailed explanation. Let $y_{thw}$ represents the ground-truth value at a give time $t$, latitude $h$ and longitude $w$, and $u_{thw}$ is the corresponding reconstructed value, the RMSE is calculated as follow:
> $$
>     \text{RMSE} = \frac{1}{N} \sum_{t=1}^{N} \sqrt{\frac{1}{HW} \sum_{h=1}^{H} \sum_{w=1}^{W} \alpha(h) (y_{thw} - u_{thw})^2},
> $$
>
> where $\alpha(h)=cos(h)/(\frac{1}{H}\sum_{h'}cos(h'))$ is the latitude weight.
> The NRMSE is calculated by normalizing RMSE with the mean of the ground-truth values:
> $$
>     \text{NRMSE}=\frac{RMSE}{\frac{1}{NHW} \sum_{t=1}^{N} \sum_{h=1}^{H} \sum_{w=1}^{W}y_{thw}}.
> $$
>
> NRMSE quantifies the relative reconstruction error normalized by the mean nutrient concentration, providing a scale-independent measure of accuracy. We hope this clarification is helpful, and please refer to Appendix D.2 of the supplementary materials for further details.
>
> > **Q7:** Are the numbers after the $\pm$ in Table 3 standard errors? Or something else? Please clarify.
>
> **Response:** Yes, the numbers following the $\pm$ symbol in Table 3 are standard errors computed over 3 trials. We will revise the caption in the final version to explicitly state this for improved clarity.
>
> > **Q8:** Figure 4: it's not super clear what P and N in the legend are without the context in the surrounding text. Maybe these can just be written out into phosphate and nitrate for improved readability.
>
> **Response:** Thank you for your suggestion. We use abbreviations "P" for phosphate and "N" for nitrate due to space limitations. We totally agree that writing out the full terms would improve clarity.  We will update the legend accordingly in final version, where space limitations are less restrictive.
>
> > **Q9:** While lines 312 - 315 state that naive-B suffers large errors because it directly propagates the full velocity field without filtering, it seems from Figure 5 that NUTS suffers (relatively) equally for gamma tending towards -1. Could the authors comment on this?
>
> **Response:** Thank you for your insightful observation. In our experiments, the perturbation level is defined as $\delta = \gamma \mathbf{v}^\*$, where $\gamma < 0$ corresponds to removing a portion of the eddy components from the flow. For instance, setting $\gamma = -1$ yields a modified flow $\mathbf{w}^* = \bar{\mathbf{w}} + \frac{1}{\epsilon}(1 + \gamma)\mathbf{v}^* = \bar{\mathbf{w}}$, effectively eliminating the eddy-driven information. This removal leads to degraded performance across models, including NUTS, as the fine-scale variability essential for accurate reconstruction is lost.
>
> > **Q10:** The paragraph starting on Line 332 doesn't make any reference to Table 4 even though it's the results in Table 4 that are under discussion. Maybe the authors could add an explicit reference in the same way they do it in the next paragraph?
>
> **Response:** Thank you for raising this point. We agree that the reference to Table 4 should be made explicit in the paragraph discussing Obs. 5. We will revise Line 334-336 to "*Among several options, the Fourier filter achieves the lowest NRMSE on both daily (0.136) and monthly (0.151) tasks, outperforming wavelet, Gaussian, and moving average filters (Table 4a).*" and Line 339-340 to "*Lower ratios—removing more unresolved eddy components—consistently improve reconstruction accuracy, while higher ratios degrade performance (Table 4b)*", including direct reference to Table 4.
>
> > **Q11:** Table 6: it seems there is result tables for a - f but the caption only has explanations for a - e (some of which don't match their respective table, I think). Please fix.
>
> **Response:** Thank you for your suggestion. In the caption of Table  6, we merged the explanations for Tables 6(b) and 6(c) under caption (b), which may have caused confusion. We will revise the caption to "
> ...(b) Analysis of model depth in the coarse module;
> (c) Analysis of model depth in the refinement module;
> (d) Evaluation of source and conservation loss terms;
> (e) Quantification of the impact of auxiliary input variables;
> (f) Assessment of sensitivity to temporal interval length...
> " in the final version, avoiding potential confusion.
>
>
> > **Q12:** Table 6 (f): what are the units of the interval length? And is this the length of the prediction or the length of the context? Also, why are the short intervals worse than the intermediate ones? This feels counterintuitive.
>
> **Response:** Thank you for your thoughtful question. The interval length is measured in days for the Daily dataset and in months for the Monthly dataset. This interval length represents the length of the context used by the model. We observe performance peaks at a moderate interval length. Short contexts provide insufficient temporal information for the initializer to capture temporal dependencies, while long contexts introduce noise and uncorrelated information which may hinder model performance.
>
> > **Q13:** Line 260: change "obatained" to "obtained"
>
> **Response:** We appreciate your careful review, and we will revise the grammar in the final version.
>
> ___
>
> Thank you once again for your valuable feedback. We hope our responses address your concerns, and we would be happy to clarify further if needed. We look forward to your thoughts.

---

> > ### Comment · Reviewer_9Yz6 · 2025-08-04
> > **Thank you**
> >
> > Thank you for the detailed rebuttal. My questions have been addressed, and I maintain my current positive score.

---

> > > ### Author Response · Authors · 2025-08-05
> > > **Thank you**
> > >
> > > Thank you for your engagement in the reviewing process of our paper. We are glad to know that our rebuttal has addressed your questions.

---

### Official Review · Reviewer_9Xop · 2025-07-04

**Clarity:** 3
**Significance:** 3
**Originality:** 3
**Rating:** 4
**Confidence:** 2

**Summary:**

This paper proposes a model named NUTS pioneers a theoretically grounded, two-scale framework for eddy-robust nutrient reconstruction, achieving SOTA accuracy. While computationally heavy and surface-limited, its scale separation strategy and proven robustness offer a template for multiscale geoscientific modeling.

**Questions:**

1. The refinement module processes each time step independently, which may ignore the temporal dependencies in the eddies, and may limit the coherence of long-term reconstructions.
2. the overview of NUTS (Figure 2) makes the method look too complicated, with multiple connections between each module, and whether there is suspicion of over-modeling, which will make the work difficult to follow.
3. I carefully want to see the model complexity, computational complexity, and time complexity of total model and each module.
4. I am very interested in confirming whether the model code will be open source and to what extent.

**Ethical Concerns:**

["NO or VERY MINOR ethics concerns only"]

**Final Justification:**

Thanks to the author for the rebuttal, I will keep my positive review score.

**Limitations:**

Please check the Weaknesses and Questions

**Quality:**

3

**Strengths And Weaknesses:**

Strengths
1. This paper integrates homogenization theory (from multiscale PDEs) into deep learning for ocean sciences. The coarse module ensures stability, while the refinement recovers details without propagating errors.
2. Formal proof (Theorem 1) guarantees immunity to mesoscale perturbations—addressing a key flaw in prior data-driven/assimilation methods.
3. The experiments are implemented on simulated (MOM6) and real (WOD) data across daily/monthly scales. it achieves 79.9% NRMSE reduction for phosphate and 19.3% for nitrate vs. best baseline on real-world (WOD) data.
Weaknesses
1. The learnable source/sink module (for biological processes) lacks detailed evaluation. Its impact on long-term nutrient cycling is not quantified beyond ablation (Table 6d).
2. NUTS has 125.6M parameters, exceeding lightweight baselines (e.g., 4DVarNet: 0.3M). Training requires high-resolution flow data and multi-step PDE integration, limiting accessibility for resource-constrained researchers. Its model complexity, computational complexity, and time complexity are also indicators that need to be considered.
3. Performance relies heavily on Fourier filtering with strict cutoff ratios (Table 4b). Real-world flows may violate assumed scale separation, risking over-smoothing in dynamically complex regions (e.g., boundary currents).

---

> ### Author Rebuttal · Authors · 2025-07-31
>
> We sincerely thank the reviewer for the valuable feedback. Below, we provide a point-by-point response and hope the response can effectively address your concerns.
>
> ## Weaknesses
>
> > **W1:** The learnable source/sink module (for biological processes) lacks detailed evaluation. Its impact on long-term nutrient cycling is not quantified beyond ablation (Table 6d).
>
> **Response:** Thank you for your insightful comment. To quantitatively evaluate its long-term effect, we added an additional analysis during the rebuttal phase: we compute the monthly variation rate of total nutrient mass, defined as the ratio of each month's total mass to that of January in the same year, across four distinct years. As shown in **Table T1**, the refinement module maintains variation rates within $\pm 0.4$\%, adhering to the mass conservation constraint. Moreover, by introducing the source module, NUTS closely follows the ground-truth (G-T) variation trends. Since the final output of NUTS is defined as the sum of the learned source contribution and the refinement correction, this result highlights the source module’s ability to capture seasonally varying source–sink dynamics and maintain physically consistent nutrient cycling over time. Experiments are conducted on the Monthly dataset with nitrate as the target nutrient.
>
> **Table T1**: Monthly variation rates (\%) of total nutrient mass from Refinement Module, NUTS and ground-truth (G-T) across two years. The variation rate is defined as the ratio of each month’s total nutrient mass to that of January within the same year.
>
> | Year | Data       | Feb.   | Mar.   | Apr.   | May   | Jun.   | Jul.    | Aug.    | Sep.    | Oct.    | Nov.    | Dec.   |
> | ---- | ---------- | ----- | ----- | ----- | ----- | ----- | ------ | ------ | ------ | ------ | ------ | ----- |
> | 2014 | Refinement | 0.00% | 0.01% | 0.03% | 0.03% | 0.04% | 0.03%  | 0.02%  | 0.00%  | 0.00%  | 0.00%  | 0.01% |
> |      | NUTS       | 0.17% | 2.14% | 5.40% | 3.10% | 3.54% | 6.78%  | 8.59%  | 10.60% | 10.04% | 6.97%  | 2.94% |
> |      | G-T        | 0.11% | 1.38% | 2.61% | 3.31% | 4.55% | 6.60%  | 8.08%  | 7.45%  | 7.28%  | 5.09%  | 0.43% |
> | 2020 | Refinement | 0.02% | 0.02% | 0.03% | 0.03% | 0.04% | 0.04%  | 0.04%  | 0.04%  | 0.03%  | 0.04%  | 0.04% |
> |      | NUTS       | 3.59% | 5.13% | 6.03% | 8.09% | 8.09% | 12.07% | 14.38% | 15.25% | 16.11% | 12.35% | 7.84% |
> |      | G-T        | 1.00% | 2.99% | 4.99% | 6.57% | 7.94% | 10.67% | 13.07% | 14.08% | 13.36% | 11.22% | 7.69% |
>
> > **W2:** NUTS contains 125.6M parameters compared to 0.3M in 4DVarNet, leading to higher model, computational, and time complexity.
>
> **Response:**  Thank you for raising this concern. While NUTS has 125.6M parameters—significantly larger than 4DVarNet (0.3M)—it achieves a markedly lower reconstruction error on phosphate (NRMSE: 0.035 $\pm$ 0.002) and demonstrates comparable computational efficiency under matched hardware settings. To ensure fair comparison, we report training and inference times in Appendix D.6 of the supplementary materials (Table 15), where all models were trained and evaluated on two RTX 6000 Ada GPUs with identical batch size (8). Under this setup, NUTS trains in **92s/epoch** and infers in **58s/epoch**, closely matching 4DVarNet's training and inference times (**94s/epoch** and **46s/epoch**, respectively), despite having over **400 $\times$ more** parameters.
>
> This underscores a key point: **efficiency is not solely determined by parameter size**. 4DVarNet incurs high cost due to its iterative variational optimization, requiring repeated forward–adjoint PDE solves and offering limited parallelism. In contrast, NUTS employs batched, multi-step PDE rollouts with data-driven initialization, enabling efficient GPU acceleration and scalable inference.
>
> Furthermore, compared to foundation-scale models like Prithvi (2.3B) and AtmoRep (0.7B), NUTS is **10-20 $\times$ smaller**, faster to train and deploy, and more accurate. It thus strikes a practical balance: accurate and efficient relative to both lightweight and large-scale baselines. Pretrained checkpoints and evaluation scripts are provided to support ease of use.
>
>
> > **W3:** Fourier filtering with strict cutoff ratio may lead to over-smoothing in dynamically complex regions such as boundary currents.
>
> **Response:** We thank the reviewer for this thoughtful observation. While Fourier filtering may suppress high-frequency features in complex regions such as boundary currents, NUTS explicitly addresses this by reintroducing the **normalized filtered component** in the refinement module to recover eddy-driven variability. To ensure physical coherence, we further impose a **mass conservation loss** during refinement, preventing artificial changes in total tracer mass. This design allows NUTS to restore fine-scale detail while maintaining global consistency.
>
> ## Questions
> > **Q1:** The refinement module may overlook temporal eddy dependencies, limiting long-term coherence.
>
> **Response:** We thank the reviewer for this valuable point. Although the refinement module operates independently at each time step, we impose a **mass conservation loss** (Eq. 162–163) to ensure no artificial tracer mass is introduced. Since the homogenized PDE preserves mass at the coarse level, this constraint enforces consistency during refinement and introduces an **implicit temporal linkage** across the refined predictions.
>
> We agree that capturing temporal dependencies in the eddy component could be beneficial. To test this, during the rebuttal phase, we implemented a spatio-temporal transformer that integrates 4 frames of eddy components into the refinement stage. However, performance worsened—NRMSE on nitrate increased from 0.151 to 0.162 on the Monthly dataset—and computation rose from 990 to 1215 GFLOPs. We suspect accumulated truncation errors in the coarse fields disrupt temporal attention, offsetting potential gains.
>
> > **Q2.1:** Figure 2 makes NUTS appear overly complex, with many inter-module connections that may hinder readability.
>
> **Response:** Thank you for raising this point. In the final version, we will simplify the figure to enhance readability. Specifically, we will merge functionally related components--for example, merging the two "**Coarse Nutrient Field**" boxes into a single box labeled "**Coarse Nutrient Field  $\bar{\varphi}(\mathbf{x}_k, t)$**". Additionally, we will streamline the data flow by removing some elements, such as the "**Mass Conservation Loss**" box, to reduce visual clutter and overlapping connections.
>
> > **Q2.2:** Is there over-modeling in NUTS?
>
> **Response:** We thank the reviewer for this thoughtful question. We believe each component in NUTS is functionally necessary, as confirmed by extensive ablation studies. (1) In the coarse module, spatio-temporal architectures outperform static baselines, and transformer-based models consistently surpass CNNs due to superior capacity (Table 6(a)). (2) In the refinement module, a moderately deep network is essential to recover fine-scale nutrient structure from eddy flows (Figure 5, Table 6(c)). (3) The source module and conservation loss both contribute measurable gains (Table 6(d)), with the conservation loss further enforcing mass conservation (Appendix E.2 of the supplementary materials). These empirical results underscore the necessity of the proposed model components.
>
> > **Q3:** I would like to see the model, computational, and time complexity of the full model and each module.
>
> **Response:** We thank the reviewer for highlighting this important point. **Table T2** provides a detailed breakdown of **parameter count**, **computation (GFLOPs)**, and **runtime per forward pass** (batch size = 8) for each component of NUTS. While the PDE solver has negligible parameter count (0.04M), its compute (181 GFLOPs) and runtime (75ms) are comparable to the coarse and refinement modules due to the cost of multi-step integration. The coarse and refinement modules account for most of the complexity (328 and 300 GFLOPs), yielding a total of 125.6M parameters, 990 GFLOPs, and 413ms per batch. We also refer the reviewer to Appendix D.6 of the supplementary materials (Table 15) for full comparisons against other models. Despite its larger size, NUTS achieves state-of-the-art accuracy while maintaining competitive training and inference efficiency under matched hardware settings.
>
> **Table T2:** Number of parameters, computation complexity and time complexity of NUTS and its components.
>
> | Module            | # Params (M) | Computation (GFLOPs) | Time (ms) |
> | :-----------------: | :------------: | :--------------------: | :---------: |
> | Coarse Module     | 83.2         | 328                  | 105       |
> | PDE Solver        | 0.04         | 181                  | 75        |
> | Refinement Module | 42.1         | 300                  | 207       |
> | Source Model      | 0.04         | 181                  | 26        |
> | **NUTS**   | **125.6**    | **990**              | **413**   |
>
> > **Q4:** Will the code be open-sourced, and if so, to what extent?
>
> **Response:** Thank you for raising this point. We open-sourced the code for training and testing our NUTS, as well as the Daily and Monthly datasets. While we are prohibited to provide external links according to the guidelines, the URL in Sec 4.1 directs to the anonymous code repository containing the code and datasets.
>
> ___
> Thanks again for your valuable reviews and we hope our response can resolve your concerns. Please don't hesitate to contact us if you have further questions. Looking forward to hearing from you.

---

> > ### Comment · Reviewer_9Xop · 2025-08-07
> > **Reply to author's rebuttal**
> >
> > Thank you for the author's rebuttal. Although a lot of supplementary experiments and comparisons are provided, it still does not dispel my doubts. Whether the performance of the model comes from large parameters or method gains. If authors can perform the experiments related to scaling, I believe it will be a better work.

---

> > > ### Author Response · Authors · 2025-08-08
> > >
> > > > **Q:** Add experiments related to scaling.
> > >
> > > **Response:** Thank you for the feedback. To address the concern on whether NUTS’ performance gains stem from large parameter counts or architectural design, we conducted controlled scaling experiments for NUTS, U-Net, and ViT, each in Small (S), Medium (M), and Large (L) configurations (**Table T1**). Results on the Monthly dataset (**Table T2**) lead to three clear observations:
> > >
> > > 1. **At matched parameter counts, NUTS is consistently better.** For example, NUTS-S (37.2M params) outperforms U-Net-S (31.0M) and ViT-S (39.9M), and NUTS-M/L similarly surpass their size-matched counterparts.
> > >
> > > 2. **Scaling improves all models.** Larger variants reduce NRMSE across U-Net, ViT, and NUTS, confirming that model size contributes positively to performance.
> > >
> > > 3. **Size alone cannot close the gap.** Even the largest baseline (U-Net-L, 150.6M) is matched or exceeded by much smaller NUTS variants, underscoring that the two-scale design is the key driver of NUTS’ advantage, not just parameter count.
> > >
> > > In short, scaling helps, but structure wins--NUTS’ two-scale design delivers consistent gains that parameter scaling alone cannot replicate. We hope these results clearly address the concern and we look forward to constructive discussions on further improving the work. We will include this scaling analysis in the final version of the paper.
> > >
> > > **Table T1:** **Model Hyperparameters.**
> > > **(a)** Hyperparameter of the three variants of U-Net, $C_{\text{Enc}}$ denote the latent channels in each encoder block;
> > > **(b)** Hyperparameter of the three variants of ViT, $L$ represents the number of ViT blocks;
> > > **(c)** Hyperparameter of the three variants of NUTS, $L_{\text{coarse}}$ and $L_{\text{refine}}$ are the depth of the coarse module and the depth of the refinement module, respectively.
> > >
> > > **(a) U-Net Hyperparameters**
> > > | Model | $C_{\text{Enc}}$       |
> > > | :-------------------------: | :-------------------------: |
> > > | U-Net (S)                 | 64, 128, 256, 512, 1024   |
> > > | U-Net (M)                 | 96, 192, 384, 768, 1536   |
> > > | U-Net (L)                 | 128, 256, 512, 1024, 2048 |
> > >
> > > **(b) ViT Hyperparameters**
> > > | Model | $L$ |
> > > | :-----------------------: | :-----: |
> > > | ViT (S)                 | 6     |
> > > | ViT (M)                 | 12    |
> > > | ViT (L)                 | 16    |
> > >
> > > **(c) NUTS Hyperparameters**
> > > | Model | $L_{\text{coarse}}$ | $L_{\text{refine}}$ |
> > > | :------------------------: | :----------------------: | :----------------------: |
> > > | NUTS (S)                 | 3                      | 1                      |
> > > | NUTS (M)                 | 6                      | 3                      |
> > > | NUTS (L)                 | 12                     | 6                      |
> > >
> > >
> > >
> > > **Table T2:** NRMSE ($\downarrow$), parameters and efficiency comparison of different models on the Monthly dataset, with nitrate as the target variable. $^*$ indicates the model variant used in the submitted paper.
> > >
> > > | Size | Model         | Number of Params (M) | NRMSE |
> > > | :-------------: | :-------------: | :-----------: | :----: |
> > > Small | $^*$ U-Net (S) |         31.0 | 0.187 |
> > > | | ViT (S)       |         39.9 | 0.270 |
> > > | | NUTS (S)      |         37.2 | **0.173** |
> > > Medium | U-Net (M)     |         84.7 | 0.175 |
> > > | | $^*$ ViT (M)   |         77.7 | 0.260 |
> > > | | NUTS (M)      |         68.7 | **0.160** |
> > > Large | U-Net (L)     |        150.6 | 0.171 |
> > > | | ViT (L)       |        102.9 | 0.249 |
> > > | | $^*$ NUTS (L)  |        125.6 | **0.151** |

---

> ### Author Response · Authors · 2025-08-05
> **Kind Reminder**
>
> Dear Reviewer 9Xop,
>
> We hope this message finds you well. As the discussion period is nearing its end, we want to ensure that we have addressed all your concerns satisfactorily. If there are any additional questions or feedback you'd like us to consider, please let us know. Your comments are valuable to improve our work.
>
> Thank you for your time and effort in reviewing our paper.

---

### Decision · Program_Chairs · 2025-09-17

**Decision:**

Accept (poster)

**Comment:**

This paper proposes a novel two-scale model (NUTS) for reconstructing ocean surface nutrients, effectively handling mesoscale eddy variability and outperforming baselines on key metrics. Results were validated on both simulated and real-world data. Reviewers agreed that not only were the theoretical principles well justified, the empirical results were also strong with good baselines and ablations. There were a few concerns w.r.t. model complexity and uncertainty quantification, which were mostly resolved during the discussion period.